

# Analyzing the interpretative ability of landscape pattern to explain thermal environmental effects in the Beijing-Tianjin-Hebei urban agglomeration

Dongchuan Wang[1,2], Zhichao Sun[1], Junhe Chen[1], Xiao Wang[1], Xian Zhang[1] and Wei Zhang[1]

[1] School of Geology and Geomatics, Tianjin Chengjian University, Tianjin, China
[2] Tianjin Key Laboratory of Civil Structure Protection and Reinforcement, Tianjin, China

## ABSTRACT

The development of the urban agglomeration has caused drastic changes in landscape pattern and increased anthropogenic heat emission and lead to the urban heat island (UHI) effect more serious. Therefore, understanding the interpretation ability of landscape pattern on the thermal environment has gradually become an important focus. In the study, the spatial heterogeneity of the surface temperature was analyzed using the hot-spot analysis method which was improved by changing the calculation of space weight. Then the interpretation ability of a single landscape and a combination of landscapes to explain surface temperature was explored using the Pearson correlation coefficient and ordinary least squares regression from different spatial levels, and the spatial heterogeneity of the interpretation ability was explored using geographical weighted regression under the optimal granularity ($5 \times 5$ km). The results showed that: (1) The hot spots of surface temperature were distributed mainly in the plains and on the southeast hills, where the landscapes primarily include artificial landscape (ArtLS) and farmland landscape (FarmLS). The cold spots were distributed mainly in the northern hills, which are dominated by forest landscape (ForLS). (2) On the whole, the interpretative ability of ForLS, FarmLS, ArtLS, green space landscape pattern, and ecological landscape pattern to explain surface temperature was stronger, whereas the interpretative ability of grassland landscape and wetland landscape to explain surface temperature was weaker. The interpretation ability of landscape pattern to explain surface temperature was obviously different in different areas. Specifically, the ability was stronger in the hills than in the plain and plateau. The results are intended to provide a scientific basis for adjusting landscape structural, optimizing landscape patterns, alleviating the UHI effect, and coordinating the balance among cities within the urban agglomeration.

Corresponding author
Dongchuan Wang,
wangdongchuan@tcu.edu.cn

## INTRODUCTION

In recent years, because of the rapid development of urbanization, artificial landscape (ArtLS) has gradually eroded the natural landscape (*Su et al., 2012*; *Angel et al., 2011*; *Miao et al., 2011*). Impermeable layers with high thermal conductivity which are composed of cement and asphalt continue to expand, and energy and material consumption keep increasing which lead to greenhouse gas and anthropogenic heat emissions increasing (*Bonafoni et al., 2017*; *Barrington-Leigh & Millard-Ball, 2015*). This has changed the heat flow and heat balance between atmosphere and land surface, aggravated the heat island effect, and has seriously threatened the urban living environment and the quality of life for dwellers (*Zhang et al., 2009*; *Zhijia et al., 2016*). Therefore, the search for mitigation methods of urban heat island (UHI) effect has become a hot research topic among scholars (*Connors, Galletti & Chow, 2013*).

Most previous studies have looked for ways to mitigate the heat island effect from its causes and have shown that factors, such as surface parameters, human activities, altitude, and average wind speed, are significantly correlated with the urban thermal pattern (*Amiri et al., 2009*; *Du et al., 2016*; *Jiao et al., 2019*). Many studies have shown that there is a significant correlation between the normalized difference vegetation index (NDVI), the normalized difference built-up index and the normalized difference water index and surface temperature, and the surface parameters are the main cause of heat island problems (*Zhao et al., 2018a*; *Xiao & Weng, 2007*; *Chen et al., 2006*), which are closely related to landscape patterns of the underlying surface (*Gage & Cooper, 2017*; *Zullo et al., 2019*). Therefore, the study on the coupling relationship between landscape pattern and surface temperature is the breakthrough to alleviate the UHI effect (*Li et al., 2011*, *2017*).

Since the landscape pattern index can accurately reflect the spatial pattern information of the landscape (*Zhao et al., 2018a*, *2018b*), the Pearson correlation coefficient and ordinary least squares (OLS) regression method can reflect the relationship between different factors (*Yi, Hu & Li, 2018*; *Asgarian, Amiri & Sakieh, 2015*; *Connors, Galletti & Chow, 2013*). Therefore, following these methods, more and more studies had been conducted to explore the relationship between landscape pattern and surface temperature. These studies have shown that the correlation between landscape area percentage index (PLAND) and surface temperature was more significant than patch density, edge density, landscape shape index, and the relationship between the PLAND of different landscapes and surface temperatures also was different (*Chen, Sun & Chen, 2012*). For example, increasing the area of forest landscape (ForLS) and wetland landscape (WetLS) could achieve a cooling effect (*Kumar & Shekhar, 2015*; *Xiao et al., 2018*), and increasing the area of ArtLS could lead to an increase in surface temperature (*Jones et al., 1990*). In addition, the relationship between the PLAND of same landscape and surface temperature varied in different regions and scales (*Shen et al., 2015*). This difference was related not only to the heterogeneity of the landscape pattern but also to the granularity of the selected analytical window (*Guo et al., 2012*; *Lu et al., 2018*; *Estoque, Murayama & Myint, 2017*). However, the current study on the interpretation ability of integrated landscapes such as ecological land (composed of forestland, grassland, and wetland) to explain surface

temperature is insufficient. In addition, some progress has been made in exploring the spatial heterogeneity of the interpretation ability of landscape pattern to explain surface temperature, but the research revealed the causes of the spatial heterogeneity from the comprehensively perspective including landscape, topography, landform, elevation difference, microclimate difference, and human activities need greater progress (*Zhou & Wang, 2011*). When exploring the heterogeneity of the relationship between the two variables, the OLS is not applicable to reflect the spatial heterogeneity of the relationship between the dependent variable and independent variables space difference, geographically weighted regression (GWR) is a kind of effective method to reflect the spatial heterogeneity, and *Li et al. (2010)* found that GWR than OLS provides a better fitting and more localized information when researching the landscape driving factors of land surface temperature (LST).

At present, most studies had made significant progress in clarifying the thermal environment problems of single cities or of a few large cities, such as Beijing and Shanghai (*Wang et al., 2017*). With the development and integration of the city, in the Beijing-Tianjin-Hebei urban agglomeration, Yangtze River Delta urban agglomeration, Pearl River Delta urban agglomeration, and other large-scale regions, the research on the thermal environment should deserve further attention (*Zhou et al., 2018*). As the main form of promoting urbanization in China, urban agglomeration has a close spatial structure and close economic ties. And as the distance between cities decreased, high-temperature regions continuously integrated and expanded, which contributed to the UHI effect becoming increasingly serious. In addition, at the 18th National Congress of the Communist Party of China and in the coordinated development strategy of Beijing-Tianjin-Hebei, it was emphasized that development of urban agglomerations is considered to be comprehensive, healthy, coordinated, and sustainable development. Thus, we need to pay more attention to the problem of the thermal environment in the Beijing-Tianjin-Hebei urban agglomeration.

From the perspective of sustainable and integrated development in Beijing-Tianjin-Hebei urban agglomeration integration, we combined data for surface temperature and NDVI in summer 2015, land cover, digital elevation model (DEM), and topography. In the paper, the determination of Euclidean distance was improved from two to three dimensions, which was included in the hot-spot analysis method, and the spatial pattern information of landscapes and surface temperatures were analyzed quantitatively. To explore the optimal analytical window granularity of the relationship between landscape pattern and surface temperature on for an urban agglomeration, we applied the Pearson correlation coefficient. Then, we combined OLS and GWR models to explain the influence and the interpretative ability of landscape pattern to explain surface temperature and their spatial heterogeneity from different spatial levels. Finally, we revealed the origin of the spatial heterogeneity from the landscapes of the basement, topography, elevation, and human activities. These results provide a scientific basis for the adjustment of land cover structure and a reference to improve the thermal environment in the Beijing-Tianjin-Hebei urban agglomeration.

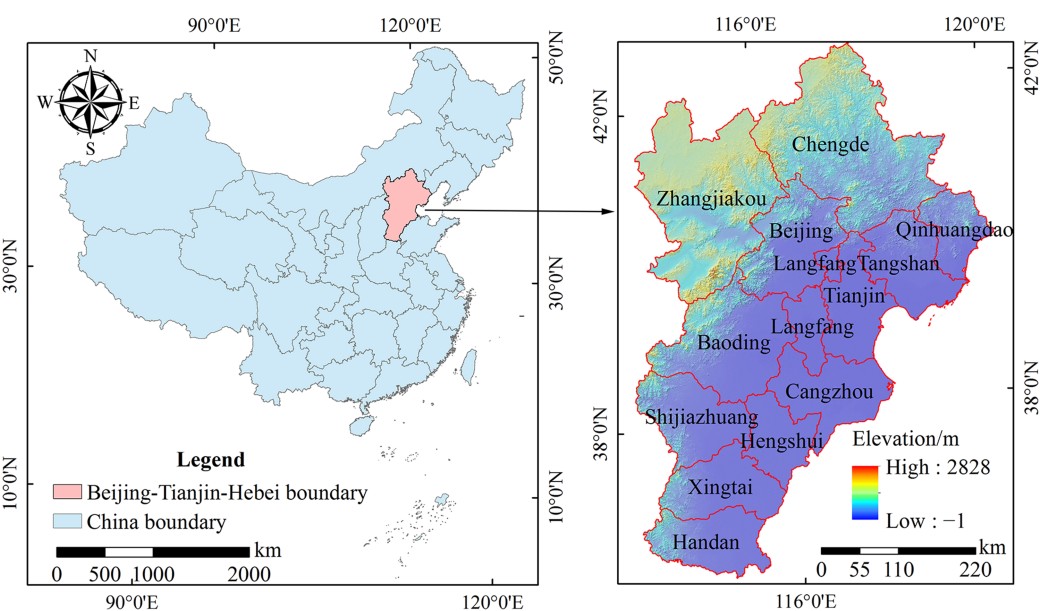

**Figure 1 Location of the Beijing-Tianjin-Hebei urban agglomeration.** In the figure, the blue areas represent the administrative areas of China, and the pink areas represent the administrative areas of the Beijing-Tianjin-Hebei urban agglomeration. A hillshaded relief (Azimuth: 315, Altitude: 45, the effect was the position of the sun in the afternoon of Beijing winter, and the transparency of this layer was 30% when the map was created) was superimposed with Elevation in the right figure.

# METHODOLOGY

## Study area

The Beijing-Tianjin-Hebei urban agglomeration is composed of Beijing, Tianjin, and 11 cities in Hebei Province (Fig. 1), which is located in north latitude 36°05′–42°40′, the area is 216,485.2 km$^2$, the area of land for construction is about 25,000 km$^2$, accounting for 11.6% which is significantly higher than the national average. According to the data of The Sixth National Census, the permanent population was 104 million in 2010, with a population density of 484 inhab/km$^2$. The city with the highest population density is Beijing, with 1,167 inhab/km$^2$. The city with the lowest population density is Hebei province, with 381 inhab/km$^2$.

The terrain is complex and the difference in elevation is significant. The terrain includes hills, plains, intermontane basins, and other types of terrain; plains and hills are the primary types of terrain. The plains are dominated by the northern China plain, which is concentrated in the eastern part of the Beijing-Tianjin-Hebei urban agglomeration. Hills are found primarily in the northwest region of the Beijing-Tianjin-Hebei urban agglomeration (Zhangjiakou and Chengde) and in the north of Baoding city, which mainly includes the Taihang, Yanshan, and Jundu mountains, and forest is the main landscape. The climate is a temperate continental climate, with severe cold in the winter and heat in the summer. The annual temperature is quite varied, so the vegetation type is primarily temperate deciduous broad-leaved forest, which is affected significantly by the climate (*Li et al., 2017*; *Miao et al., 2015*; *Zhang et al., 2017*).

**Table 1 Data sources.** The name, source, date, type, resolution and download website of the data are shown.

| Name | Source | Date | Type | Resolution/m | Website |
|---|---|---|---|---|---|
| Land cover | Landsat8 OLI remote sensing image | July 12, 2015, July 28, 2015 | Raster | 30 | Geospatial data cloud (http://www.gscloud.cn/) |
| DEM | ASTER GDEM remote sensing image | | Raster | 30 | |
| Surface temperature | MODIS surface temperature 8 days synthesis product (MOD 11A2) | July 4, 2015, July 12, 2015 July 20, 2015, July 28, 2015 | Raster | 1,000 | EARTHDATA Search (https://search.earthdata.nasa.gov/search) |
| NDVI | MODIS vegetation index 16 days synthetic product (MOD 13A1) | July 12, 2015, July 28, 2015 | Raster | 500 | |
| Topographic | | | Vector | | Geographic Information Monitoring Cloud Platform (http://www.dsac.cn/) |

## Data source and preprocessing

The data used in this study mainly included land cover, DEM, surface temperature, topography and NDVI data for the Beijing-Tianjin-Hebei urban agglomeration (Table 1). We obtained the land cover data from the Landsat8 OLI remote-sensing images (2015) with cloud cover less than 5% after manual visual interpretation. The resolution of the land cover data is $30 \times 30$ m. The land cover types that we interpreted included forest, grassland, wetland, farmland, artificial surface, and bare land. We verified the accuracy of the interpretation using historical data and field investigation during the same period, and the accuracy was higher than 93%.

The northern and southern latitude span is large in the Beijing-Tianjin-Hebei urban agglomeration. To reduce the influence of latitude position, seasonal change, and cloud cover on the accuracy of the relationship between landscape pattern and surface temperature, we selected surface temperature synthetic products (MOD11A2) in four phases (July 4, July 12, July 20, July 28, 2015) and vegetation index synthetic products (MOD13A1) in two phases (July 12 and July 28, 2015) which always were used to reflect surface temperature and NDVI in larger areas (*Ferreira et al., 2013*; *Chen et al., 2011*). First of all, we extracted "LST_Day_1km" and "NDVI" from the data of two products using ENVI 5.3. Then, we preprocessed and averaged these products. Finally, we obtained average surface temperature data and average NDVI data for the Beijing-Tianjin-Hebei urban agglomeration for July 2015. The resolution of the two data is $1,000 \times 1,000$, $500 \times 500$ m, respectively.

The vector data sources for topography included 10 types of topography, including alluvial fan, alluvial fan, marine plain, yellow pan plain, flood plain, depression, mountain, hill, intermontane basin, and plateau.

## Methods

The process in this study consists of five steps: firstly, classifying landscape types according to primary classification results of land cover and constructing landscape pattern;

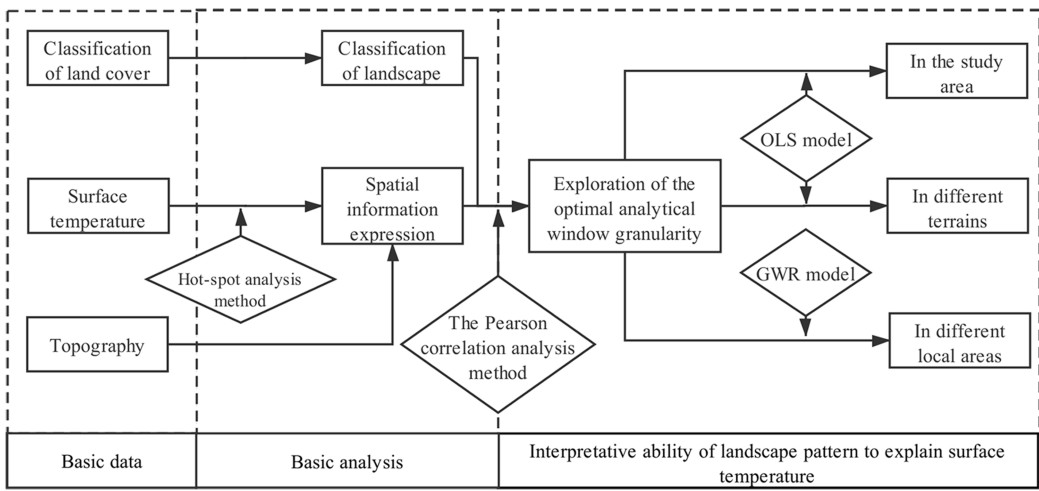

**Figure 2 The flowchart of the paper structure.** In the figure, mainly expresses the flow chart of this study, which includes the main data, main content and main research methods of this study.

secondly, dividing the study area according to terrain; thirdly, expressing the spatial information of surface temperature using the improved hot-spot analysis method; fourthly, exploring the optimal analytical window granularity of the relationship between landscape pattern and surface temperature using the Pearson correlation analysis method; finally, studying the interpretative ability of landscape pattern to explain surface temperature using OLS and GWR from different spatial levels. The flow chart of this study is shown in Fig. 2.

### Classification of landscape

Previous research has shown that the relationship between a single landscape and surface temperature has been well studied (*Zhao et al., 2018a*). In real life, however, the surface temperature is the result of the mutual influence of various landscapes. Therefore, we explored the relationship between surface temperature and two kinds of integrated landscape pattern, as well as a single landscape.

Since the change of landscape pattern is related to human activities (*Li et al., 2016*), we divided the landscapes of the Beijing-Tianjin-Hebei urban agglomeration into six landscapes according to the degree of human influence on the landscape shaping process and primary classification results of land cover: (1) ForLS, mainly referring to the forest land in the land cover classification; (2) grassland landscape (GraLS), mainly referring to grassland in the land cover classification; (3) WetLS, mainly referring to rivers, lakes, and other water areas in the land cover classification; (4) farmland landscape (FarmLS), mainly referring to cultivated land in land cover classification; (5) ArtLS, mainly referring to the artificial surface in land cover classification; (6) bare land landscape, mainly referring to the unused land in the land cover classification. Ecological landscape refers to the types of land cover mainly providing ecosystem services, including forest, garden, grassland, water area, wetland, and other natural land, such as glaciers. Green landscape generally refers to the area with green plants, including park green space, production green

space and protection green space, and so on where NDVI values tend to be high. Therefore, in this study, landscapes were integrated into two kinds of patterns, the ecological landscape pattern (EcoLSP, made up of three kinds of landscapes, ForLS, GraLS, and WetLS) and green space landscape pattern (GreenLSP), which was expressed using NDVI. In the study, we used all the single landscapes and integrated landscape patterns to explore the relationship between landscape pattern and surface temperature using the Pearson correlation analysis method and OLS regression method.

### Division of study area according to terrain

In mountainous areas with complex environmental conditions, key topographic factors, such as altitude and slope, have a significant impact on surface temperature (*Jiao et al., 2019*; *Duan et al., 2015*). The terrain in the Beijing-Tianjin-Hebei urban agglomeration is complex, including hills that include the Taihang, Yanshan, and Jundu mountains and parts of the northern China plain, with significant differences in elevation between these different types of terrain.

    To explore the influence of altitude on surface temperature, based on the grid of $1 \times 1$ km, we statistically analyzed the average surface temperature and average DEM values in different grids. We found a significant correlation between altitude and surface temperature using the Pearson correlation analysis method (the correlation coefficient was $-0.68^{**}$). To reduce the influence of topographic factors on the relationship between landscape pattern and surface temperature, we divided the region in the Beijing-Tianjin-Hebei urban agglomeration into three types of terrain—plain, hills, and plateau (Fig. 3).

### Spatial information expression of surface temperature

The *Getis-Ord $G_i^*$* local hot-spot analysis method can be used to identify the spatial clustering regions with high value and low value of statistical significance. Therefore, in this part, we used this method to carry out clustering analysis and spatial heterogeneity expression of surface temperature. We combined the results of the hot-spot analysis with the landscape pattern and terrain to reveal landscape composition information for hot and cold spots in different types of terrain in this region. The formula of *Getis-Ord $G_i^*$* (*Feng et al., 2018*) was as follows:

$$G_i^* = \frac{\sum_{j=1}^{n} w_{ij}x_j - \bar{x}\sum_{j=1}^{n} w_{ij}}{\sqrt{\frac{\sum_{j=1}^{n} x_j^2}{n} - x^2}\sqrt{\frac{\left[n\sum_{j=1}^{n} w_{ij}^2 - \left(\sum_{j=1}^{n} w_{ij}\right)^2\right]}{n-1}}}, \tag{1}$$

where $n$ represents the total number of factors in the study area; $w_{ij}$ is the spatial weight coefficient between the factor $i$ and factor $j$, reflecting the spatial relationship; $x_j$ is the attribute value of the factor $j$; and $\bar{x}$ is the average attribute value of $n$ factors.

    The method to determine the spatial weight coefficient is the key to determining the degree of influence between two factors. In this part, we selected the inverse distance weighting method to calculate the spatial weight coefficient. The traditional inverse distance weighting method was constructed primarily in the two-dimensional coordinate

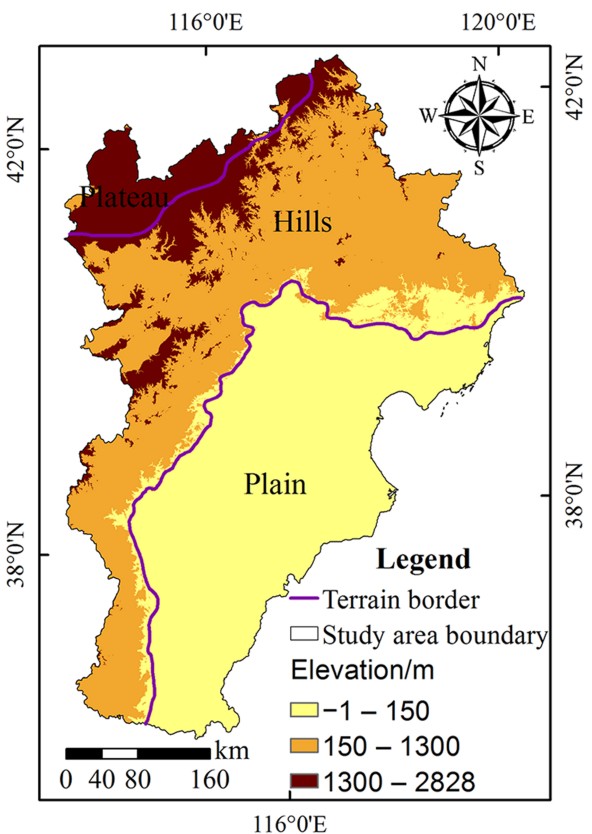

**Figure 3 Terrains and elevation of the Beijing-Tianjin-Hebei urban agglomeration.** In the figure, the purple line refers to the dividing line between different terrains, the different colors represent different elevation values. The gradient color legend and the classification of elevations distinguish the terrains including plains, hills, and plateau.

system. In hills, however, altitude has a significant impact on the Euclidean distance between two points. Therefore, we combined DEM data and the two-dimensional coordinate system to construct the three-dimensional coordinate system to calculate the distance and weight in the three-dimensional coordinate system. The formula of $w_{ij}$ follows:

$$w_{ij} = \frac{\left((X_j - X_i)^2 + (Y_j - Y_i)^2 + (Z_j - Z_i)^2\right)^{-\frac{p}{2}}}{\sum_{a=1}^{m}\left((X_j - X_i)^2 + (Y_j - Y_i)^2 + (Z_j - Z_i)^2\right)^{-\frac{p}{2}}} \quad (i \neq j),$$

(2)

where $m$ represents the total number of factors in the local neighborhood of factor $i$, and it is 8 according to the best practice criterion; $X_j$, $Y_j$, and $Z_j$ are the X, Y, and Z coordinate values of the factor $j$; $X_i$, $Y_i$, and $Z_i$ are the X, Y, and Z coordinate values of the factor $i$; and $p$ is the parameter, which is determined by the minimum value of the root mean square error, where it is 2. To avoid the situation in which the distance between two elements is 0, $w_{ij} = 1$ when $i = j$.

**Table 2 Z score, p-value, confidence level, and explanation.**

| Z score | p | Confidence level | Explanation |
|---|---|---|---|
| $Z(G_i^*) < -1.65$ | <0.1 | >90% | Significant cold spots |
| $-1.65 < Z(G_i^*) < 1.65$ | >0.1 | <90% | Insignificant region |
| $Z(G_i^*) > +1.65$ | <0.1 | >90% | Significant hot spots |

**Note:**
The results of hot-spot analysis are illustrated, such as Z score, p-value, confidence level, and explanation.

$Z$ score, $p$ values, and confidence level can reflect a statistically significant aggregation or discrete model of factors or the value. The $z$-score $(Z(G_i^*))$ is the statistic of $G_i^*$, which is a multiple of the standard deviation; $p$ is the probability. The relationship among the three is shown in Table 2, and the relevant contents in the table refer to the help document of ArcGIS 10.2 software. The formula of $Z(G_i^*)$ is as follows:

$$Z(G_i^*) = \frac{1 - E(G_i^*)}{\sqrt{\text{VAR}(G_i^*)}}, \tag{3}$$

where $E(G_i^*)$ is the expected value of the local exponent of *Getis-Ord* $G_i^*$; and $\text{VAR}(G_i^*)$ is the variance of the local exponent of *Getis-Ord* $G_i^*$.

### Exploration of the optimal analytical window granularity of the relationship between landscape pattern and surface temperature

The relationship between surface temperature and landscape pattern is correlated significantly with the analytical window granularity. If the analytical window granularity is too small, the calculation will be too large, the information of landscape pattern inside the window will be single, and the excessive analytical window granularity will affect the full embodiment of heterogeneity (*Estoque, Murayama & Myint, 2017*). Therefore, based on the resolution of surface temperature data (1 × 1 km), we selected eight kinds of analytical window granularity, including 1 × 1, 3 × 3, 5 × 5, 7 × 7, 9 × 9, 11 × 11, 13 × 13, and 15 × 15 km, and explored the optimal analytical window granularity of the relationship between landscape pattern and surface temperature in the Beijing-Tianjin-Hebei urban agglomeration.

Following the principle of grid statistical analysis, we divided the research area into multiple grids. On the basis of this principle, we then analyzed the average temperature, the PLAND of different landscapes and the average NDVI for each grid. Therefore, the phenomenon that the resolution of basic data was not identical has little effect on our research results.

We used the Pearson correlation analysis method to express the relationship between two factors by observing the correlation coefficient $r$ (*Lu et al., 2018*; *Miao, Ni & Borthwick, 2010*). We determined the analytical window granularity by comparing $r^2$.

### Interpretative ability of landscape pattern to explain surface temperature

On the basis of the optimal analytical window granularity, we combined OLS and GWR model to explore the interpretative ability and degree of influence of landscape pattern on surface temperature from different spatial levels.

The OLS model reflected the positive and negative relationship between the interpretative variables and explained the variable and the strength of the relationship by the regression coefficient ($\beta$). The $R^2$ of the results reflected the fitting effect of the evaluation model. The formula of the regression model and $R^2$ (*Farahani et al., 2010*) is as follows:

$$y_i = \beta_0 + \beta x_i + \varepsilon_i, \quad \text{and} \tag{4}$$

$$R^2 = \left( \frac{\sum_{i=1}^{n}(x_i - \bar{x})(y_i - \bar{y})}{\sqrt{\sum_{i=1}^{n}(x_i - \bar{x})^2}\sqrt{\sum_{i=1}^{n}(y_i - \bar{y})^2}} \right)^2. \tag{5}$$

In formula (4), $y_i$ represents the explained variable value of the sampling point $i$; $x_i$ represents the interpretative variable value of the sampling point $i$; $\beta_0$ is the intercept, and $\beta$ is the slope or regression coefficient of the interpretative variable; and $\varepsilon_i$ is the residual. In formula (5), the means of $x_i$ and $y_i$ are the same as the means in formula (4), $\bar{x}$ represents the average value of the interpretative variable, $\bar{y}$ represents the average value of the explained variable, and $n$ represents the total number of explained variables.

Although the OLS model can express the interpretative ability of landscape pattern to explain surface temperature as a whole, it cannot accurately reflect the spatial heterogeneity. On the basis of OLS model and GWR model, we incorporated spatial characteristics of the data and used subsamples from nearby observations to estimate each point in the study area. The regression results reflected the spatial heterogeneity of the interpretative ability and the degree of influence. The GWR formula (*Guo, Ma & Zhang, 2008*; *Zhao et al., 2018a*) is as follows:

$$y_i = \beta_0(u_i, v_i) + \sum_{k=1}^{p} \beta_k(u_i, v_i)x_{ik} + \varepsilon_i, \tag{6}$$

where $y_i$ represents the value of the explained variable of the sampling point $i$, $x_{ik}$ represents the value of the interpretative variable $k$ at the sampling point $i$, $(u_i, v_i)$ represents the coordinates of the sampling point $i$, $\beta_0(u_i, v_i)$ represents the statistical regression constant of the sampling point $i$, $\beta_k(u_i, v_i)$ represents the regression coefficient $k$ of the sampling point $i$, its purpose is to estimate the adjacent space observations, $p$ is the number of variables participating in the regression at a sampling point, and $\varepsilon_i$ is the residual. We adopted the kernel function for the estimation function of $\beta_k(u_i, v_i)$, and the following formula of $\beta_k(u_i, v_i)$:

$$\beta_k(u_i, v_i) = \left(X^T W(u_i, v_i)X\right)^{-1} X^T W(u_i, v_i)y, \tag{7}$$

where $X$ and $X^T$ is the matrix and transpose of the factor values in the neighborhood of the observation point $(u_i, v_i)$; and $W(u_i, v_i)$ represents the distance weight matrix of the observation point $(u_i, v_i)$, which is the observation function between the observation point

**Table 3 Area ratio of different landscapes.**

| Region | | Area ratio of landscape (%) | | | | | |
|---|---|---|---|---|---|---|---|
| | | ForLS | GraLS | WetLS | FarmLS | ArtLS | BareLS |
| Study area | The whole area | 32.83 | 8.82 | 2.66 | 43.45 | 11.96 | 0.27 |
| | Hot spots | 2.81 | 7.51 | 1.74 | 62.03 | 25.70 | 0.21 |
| | Cold spots | 78.12 | 6.65 | 1.51 | 12.27 | 1.20 | 0.25 |
| Plains | The whole area | 2.80 | 0.87 | 4.93 | 69.44 | 21.82 | 0.15 |
| | Hot spots | 1.07 | 1.01 | 2.04 | 67.39 | 28.43 | 0.07 |
| | Cold spots | 0.43 | 0.99 | 40.28 | 51.21 | 6.95 | 0.14 |
| Hills | The whole area | 59.46 | 11.91 | 0.92 | 22.48 | 5.05 | 0.18 |
| | Hot spots | 9.41 | 23.87 | 0.52 | 48.95 | 16.73 | 0.51 |
| | Cold spots | 80.99 | 5.83 | 0.73 | 11.22 | 1.09 | 0.14 |
| Plateau | The whole area | 6.21 | 35.60 | 2.39 | 48.77 | 5.08 | 1.94 |
| | Hot spots | 1.81 | 52.84 | 4.20 | 32.49 | 4.52 | 4.13 |
| | Cold spots | 30.59 | 34.40 | 2.52 | 26.63 | 2.93 | 2.93 |

**Note:**
It shows the area ratio of different landscapes at different spatial levels.

and other points and is composed of factor $W_{ij}$ in the distance weight matrix. The formula of $W_{ij}$ is as follows:

$$W_{ij} = \exp\left(-\frac{d_{ij}^2}{h^2}\right), \tag{8}$$

where $h$ is the kernel bandwidth; and $d_{ij}$ is the distance between the observation point $i$ and the factors $j$ in the neighborhood. In this study, we used Akaike information criterion (AIC) to determine the optimal bandwidth, and the weight outside the bandwidth was 0. AIC was a standard used to measure the goodness of statistical model fit, which was based on the concept of entropy and provided a standard to weigh the complexity of estimation model and the goodness of fitting data. In general, when AIC was minimum, bandwidth $h$ was the best. The expression formula is as follows:

$$AIC = 2k - 2\ln(L), \tag{9}$$

where $k$ is the number of model parameters and represents the complexity of the model; and $L$ is the likelihood function, which can reflect the degree of difference between different models.

# RESULTS

## Spatial information characteristics of landscape pattern and surface temperature

Forest landscape, FarmLS, and ArtLS were the main landscapes of the Beijing-Tianjin-Hebei urban agglomeration (Table 3). Among them, ForLS was distributed primarily in hills, FarmLS was distributed mainly in the plains, and ArtLS was distributed mainly in Beijing, Tianjin, Shijiazhuang, and other plain cities, accounting for 32.83%, 43.45%, and

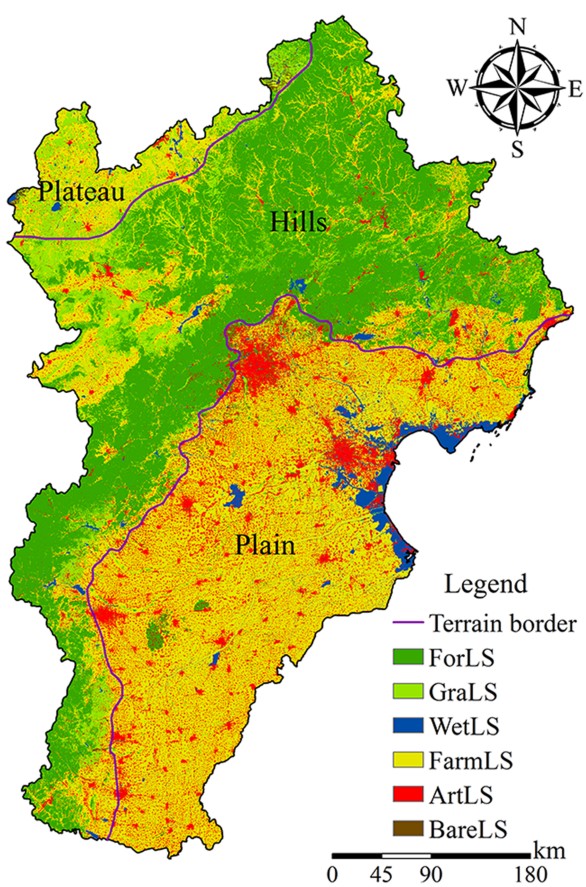

**Figure 4 Spatial distribution of landscape.** Dark green areas are ForLS, light green areas are GraLS, blue areas are WetlLS, yellow areas are FarmLS, red areas are ArtLS, and brown areas are BareLS.

11.96% of the total area, respectively (Fig. 4). Plains and hills were the main terrains in the Beijing-Tianjin-Hebei region, and the plateau was small. In the plains, FarmLS and ArtLS were the primary landscape. The area ratio of both reached 90%; in hills; ForLS, FarmLS, and GraLS were the main landscapes, with area ratios of 59.46%, 22.48%, and 11.91%, respectively. On the plateau, the landscape was dominated by FarmLS and GraLS, with an area ratio of 48.77% and 35.60%, respectively.

Hot spots of surface temperature were distributed mainly in the southwest of the plains, the southeast of the hills, and the cities like Beijing, Tianjin, Shijiazhuang; cold spots are distributed mainly in the northern hills and in Yanshan and Taihang mountains (Fig. 5). In the Beijing-Tianjin-Hebei urban agglomeration, FarmLS and ArtLS were the main landscapes of the hot spots, accounting for 62.03% and 25.70% of the total area of the hot spots, respectively. The cold spots were dominated by ForLS, accounting for 78.12% of the total cold spot area. In the plains, hot spots were dominated by FarmLS and ArtLS, whereas cold spots were dominated by FarmLS and WetLS. In hills, hot spots were dominated by FarmLS, GraLS, and ArtLS, whereas cold spots were dominated by ForLS. On the plateau, hot spots were dominated by GraLS and FarmLS, whereas cold spots were dominated by GraLS, ForLS, and FarmLS.

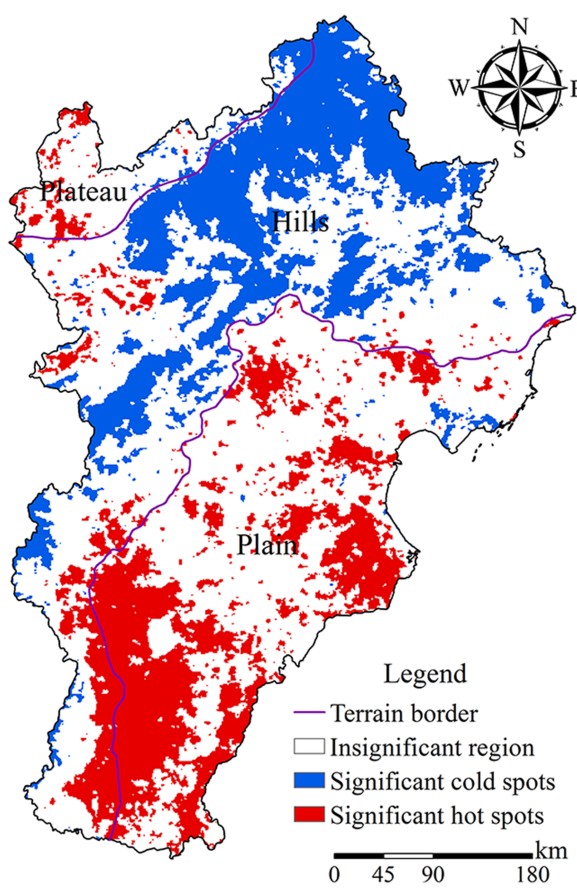

**Figure 5 Hot-spot analysis of surface temperature.** The blue area is the significant cold spot area of surface temperature, the red area is the significant hot spot area of surface temperature, and the blank area is the cold/hot spot area with no obvious concentration of surface temperature.

## Relationship between landscape pattern and surface temperature based on different analytical window granularity

The experiment showed that as window granularity increased gradually (Fig. 6), the $r^2$ of different landscape fluctuated up and down, but the fluctuation range was small. The window granularity of 5 × 5 km was the turning point at which the $r^2$ of ForLS, EcoLSP, and GreenLSP transitioned from weak to strong and the $r^2$ of the other landscapes was relatively strong. As window granularity increased gradually, the $r^2$ of FarmLS and ArtLS increased gradually. Moreover, if the analysis granularity is too large, the amount of data will be reduced, thus affecting the accuracy of the experiment. So the optimal analytical window granularity was 5 × 5 km.

## Interpretative ability of landscape pattern to explain surface temperature
### Interpretative ability of landscape pattern to explain surface temperature in the study area

According to Table 4, we found that the interpretative ability of ForLS, FarmLS, ArtLS, GreenLSP, and EcoLSP to explain surface temperature was strong, whereas the

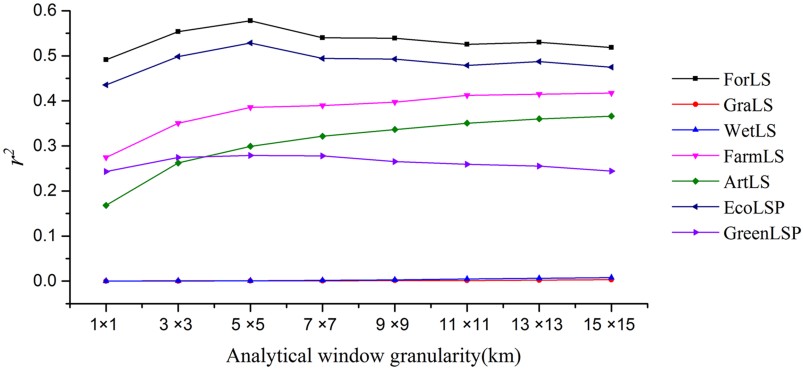

**Figure 6 The square of Pearson correlation coefficient ($r^2$) based on different analytical window granularity.** The black line represents ForLS, the red line represents GraLS, the blue line represents WetLS, the pink line represents FarmLS, the green line represents ArtLS, the dark blue line represents EcoLSP, and the purple line represents GreenLSP.

**Table 4 Analysis results of landscape's ability to interpret surface temperature in the whole domain.**

| Landscape | OLS regression | $R^2$ | $r$ |
|---|---|---|---|
| ForLS | $Y = -6.94X + 33.84$ | 0.57 | $-0.76$** |
| GraLS | $Y = -0.43X + 31.58$ | 0.00 | $-0.02$ |
| WetLS | $Y = 0.89X + 31.52$ | 0.00 | $0.03$ |
| FarmLS | $Y = 6.47X + 28.76$ | 0.39 | $0.55$** |
| ArtLS | $Y = 12.60X + 30.05$ | 0.30 | $0.62$** |
| EcoLSP | $Y = -6.34X + 34.41$ | 0.56 | $-0.73$** |
| GreenLSP | $Y = -14.53X + 41.31$ | 0.28 | $-0.53$** |

**Note:**
** There was a significant correlation at the 0.01 level (bilateral).

interpretative ability of GraLS and WetLS to explain surface temperature was weaker. The area ratio of ForLS and EcoLSP increased by 10%, the surface temperature dropped by 0.69 and 0.63 °C, respectively; the NDVI value increased by 0.1 which could represent GreenLSP, the surface temperature dropped by 1.45 °C. The area ratio of ArtLS and FarmLS increased by 10%, and the surface temperature increased by 1.26 and 0.65 °C, respectively, which indicated that ArtLS and FarmLS was the main thermal landscape, moreover, the influence of ArtLS on surface temperature is twice as great as that of FarmLS.

### Interpretative ability of landscape pattern to explain surface temperature in different terrains

The interpretative ability of ForLS, FarmLS, ArtLS, GraLS, and EcoLSP was stronger in hills (Table 5); the interpretative ability of the GreenLSP was stronger in hills and the plateau; and the interpretative ability of WetLS was stronger in the plain.

The area ratio of ForLS and EcoLSP increased by 10% and the surface temperature dropped by 0.64 and 0.63 °C in the hills and dropped by 0.45 and 0.41 °C in the plains and dropped by 0.99 and 0.18 °C in the plateau, respectively. The NDVI value increased by 0.1,

**Table 5 Analysis results of landscape's ability to interpret surface temperature after zoning.**

| Landscape | Terrain | OLS regression | $R^2$ | $r$ |
|---|---|---|---|---|
| ForLS | Plain | $Y = -4.45X + 34.20$ | 0.07 | −0.27** |
| | Hills | $Y = -6.44X + 33.45$ | 0.44 | −0.67** |
| | Plateau | $Y = -9.97X + 32.25$ | 0.42 | −0.64** |
| GraLS | Plain | $Y = -1.37X + 34.06$ | 0.03 | −0.04 |
| | Hills | $Y = 5.05X + 29.10$ | 0.08 | 0.28** |
| | Plateau | $Y = 2.19X + 30.54$ | 0.04 | 0.19* |
| WetLS | Plain | $Y = -3.87X + 34.26$ | 0.13 | −0.36** |
| | Hills | $Y = 2.44X + 29.68$ | 0.00 | 0.05 |
| | Plateau | $Y = 4.48X + 31.22$ | 0.03 | 0.17* |
| FarmLS | Plain | $Y = 1.18X + 33.29$ | 0.02 | 0.30** |
| | Hills | $Y = 6.72X + 28.15$ | 0.25 | 0.51** |
| | Plateau | $Y = 5.86X + 31.07$ | 0.02 | 0.16* |
| ArtLS | Plain | $Y = 3.19X + 33.35$ | 0.09 | 0.16* |
| | Hills | $Y = 16.87X + 28.74$ | 0.26 | 0.50** |
| | Plateau | $Y = 1.69X + 30.56$ | 0.03 | 0.14* |
| EcoLSP | Plain | $Y = -4.06X + 34.45$ | 0.21 | −0.15* |
| | Hills | $Y = -6.34X + 34.21$ | 0.34 | −0.58** |
| | Plateau | $Y = -1.78X + 32.22$ | 0.03 | −0.18* |
| GreenLSP | Plain | $Y = -1.01X + 34.69$ | 0.01 | −0.07 |
| | Hills | $Y = -22.12X + 45.34$ | 0.53 | −0.73** |
| | Plateau | $Y = -20.79X + 43.16$ | 0.68 | −0.83** |

**Notes:**
** There is a significant correlation at the 0.01 level (bilateral).
* There is a significant correlation at the 0.05 level (bilateral).

the surface temperature dropped by 2.21 °C in the hills, whereas the surface temperature dropped by 0.10 °C in the plains. It was obvious that the cooling effect of the ForLS, EcoLSP, and GreenLSP in the hills was more significant.

Artificial landscape and FarmLS were the main thermal landscapes. The area ratio of the ArtLS increased by 10%, the surface temperature rose by 1.69 °C in the hills and increased by 0.32 and 0.17 °C, respectively in the plateau and plain region. The area ratio of FarmLS increased by 10%, the surface temperature rose by 0.67 °C in the hills and rose by 0.59 and 0.12 °C, respectively in the plateau and plains. The degree of influence of ArtLS and FarmLS was more significant in the hills.

Although the interpretative ability of GraLS and WetLS to explain surface temperature was not strong in the Beijing-Tianjin-Hebei urban agglomeration, the relationship among GraLS, WetLS and surface temperature was significantly positive in the hills and the plateau, whereas it was significantly negative in the plains.

### Interpretative ability of landscape pattern to explain surface temperature using the GWR model

In the large-scale area of the Beijing-Tianjin-Hebei urban agglomeration, the area of GraLS and WetLS was small, and the relationship between GraLS, WetLS, and surface

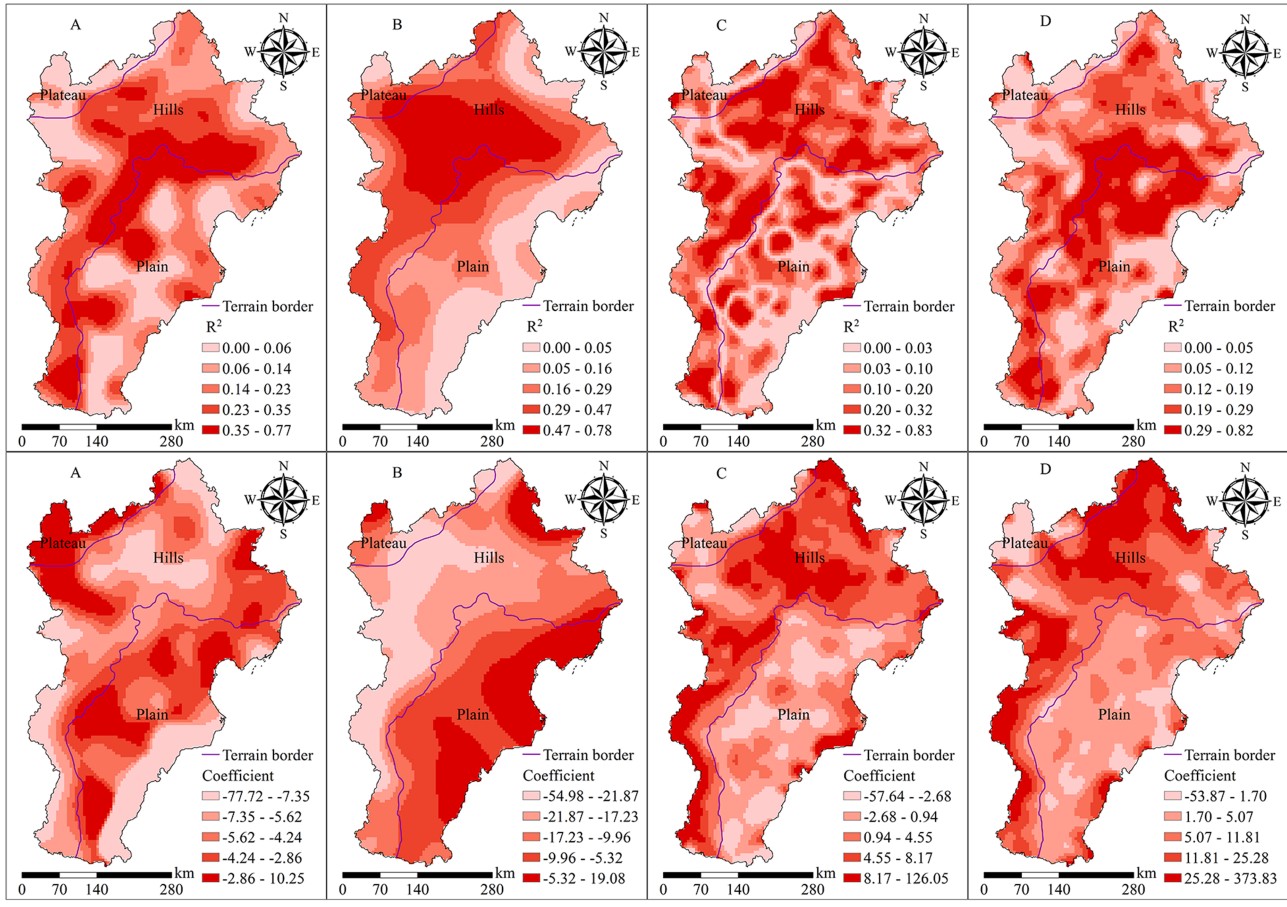

**Figure 7 Results of the relationship between landscape pattern and surface temperature ($R^2$ and regression coefficient) using GWR: (A) EcoLSP, (B) GreenLSP, (C) FarmLS, and (D) ArtLS.** The darker the color, the greater the coefficient/$R^2$.

temperature was not obvious. Therefore, we studied only the interpretative ability of the ArtLS, FarmLS, GreenLSP, and EcoLSP to explain surface temperature and we divided the research results into five grades according to the quantile method (Fig. 7).

On the whole, EcoLSP and GreenLSP had a significant cooling effect. Increasing the area of ArtLS and FarmLS could promote the increase in surface temperature (Tables 6 and 7), which was consistent with the results given in Tables 4 and 5. Spatial heterogeneity was evident, however, in the interpretative ability and degree of influence of the four landscapes on surface temperature.

In hills, the class IV and V of EcoLSP $R^2$ values accounted for 76.13% and 62.03% of the terrain, respectively, and the class IV and V coefficients accounted for 61.32% and 78.85%, respectively (Table 7). This indicated that the interpretative ability and cooling ability of EcoLSP to explain surface temperature were greater in hills than in the plains and the plateau, and the interpretative ability was stronger at the border of the plains and hills, such as in the Yanshan and Taihang mountains (Fig. 7).

In hills, the class IV and V of GreenLSP $R^2$ values accounted for 75.25% and 83.82% of the terrain, respectively, and the class I and II coefficients accounted for 88.29% and

**Table 6 GWR results of the relationship between landscape pattern and surface temperature.**

| Landscape | GWR regression coefficient | | | Average $R^2$ | Bandwidth/m |
|---|---|---|---|---|---|
| | Minimum | Maximum | Average | | |
| EcoLSP | −77.73 | 10.86 | −6.36 | 0.56 | 31,190.35 |
| GreenLSP | −54.98 | 19.61 | −12.98 | 0.53 | 46,513.73 |
| FarmLS | −61.08 | 162.27 | 3.98 | 0.65 | 16,327.41 |
| ArtLS | −55.32 | 410.73 | 16.11 | 0.62 | 18,669.30 |

Note:
GWR results of the relationship between landscape pattern and surface temperature.

**Table 7 Relationship between regression coefficient and $R^2$ classification in different terrains.**

| Landscape | Classification results | Classification standard of coefficient | Area ratio (%) | | | Classification standard of $R^2$ | The area ratio (%) | | |
|---|---|---|---|---|---|---|---|---|---|
| | | | Plain | Hills | Plateau | | Plain | Hills | Plateau |
| EcoLSP | I | −77.73 to −7.35 | 38.38 | 61.32 | 0.30 | 0.00–0.06 | 56.59 | 21.91 | 21.50 |
| | II | −7.35 to −5.62 | 20.51 | 78.85 | 0.64 | 0.06–0.14 | 46.15 | 48.03 | 5.82 |
| | III | −5.62 to −4.24 | 43.77 | 53.92 | 2.31 | 0.14–0.23 | 42.63 | 54.69 | 2.68 |
| | IV | −4.24 to −2.86 | 55.36 | 41.36 | 3.28 | 0.23–0.35 | 23.47 | 76.13 | 0.40 |
| | V | −2.86–10.25 | 48.32 | 27.63 | 24.04 | 0.35–0.77 | 37.97 | 62.03 | 0.00 |
| GreenLSP | I | −54.98 to −21.83 | 0.00 | 88.29 | 11.71 | 0.00–0.05 | 74.25 | 21.08 | 4.68 |
| | II | −21.83 to −17.23 | 7.08 | 87.17 | 5.75 | 0.05–0.16 | 62.22 | 33.35 | 4.43 |
| | III | −17.23 to −9.96 | 31.87 | 59.80 | 8.33 | 0.16–0.29 | 43.14 | 48.81 | 8.05 |
| | IV | −9.96 to −5.32 | 86.61 | 11.19 | 2.20 | 0.29–0.47 | 15.22 | 75.25 | 9.53 |
| | V | −5.32–19.08 | 81.53 | 16.10 | 2.37 | 0.47–0.78 | 12.48 | 83.82 | 3.70 |
| FarmLS | I | −57.64 to −2.68 | 80.89 | 5.82 | 13.29 | 0.00–0.03 | 67.88 | 20.06 | 12.06 |
| | II | −2.68–0.94 | 71.91 | 18.86 | 9.23 | 0.03–0.10 | 47.29 | 44.85 | 7.85 |
| | III | 0.94–4.55 | 36.15 | 59.37 | 4.48 | 0.10–0.20 | 35.52 | 59.27 | 5.21 |
| | IV | 4.55–8.17 | 11.22 | 86.40 | 2.38 | 0.20–0.32 | 31.64 | 64.74 | 3.61 |
| | V | 8.17–126.05 | 6.26 | 92.69 | 1.05 | 0.32–0.83 | 24.45 | 73.81 | 1.74 |
| ArtLS | I | −53.87–1.70 | 65.19 | 18.46 | 16.36 | 0.00–0.05 | 47.54 | 30.37 | 22.08 |
| | II | 1.70–5.07 | 81.64 | 16.18 | 2.19 | 0.05–0.12 | 38.52 | 56.21 | 5.27 |
| | III | 5.07–11.81 | 45.06 | 51.63 | 3.31 | 0.12–0.19 | 29.20 | 68.41 | 2.39 |
| | IV | 11.81–25.28 | 10.67 | 85.10 | 4.22 | 0.19–0.29 | 34.99 | 64.26 | 0.75 |
| | V | 25.28–373.87 | 3.05 | 92.56 | 4.39 | 0.29–0.82 | 56.20 | 43.52 | 0.29 |

87.17% of the terrain, respectively (Table 7). The class V of GreenLSP $R^2$ was concentrated in the central hills, including the Yanshan Mountains. This coefficient showed a trend of gradual decrease from the southeast to the northwest in space (Fig. 7), which indicated that GreenLSP had a stronger interpretative power and a greater degree of influence on the surface temperature in the hills.

In hills, the class IV and V of FarmLS $R^2$ values accounted for 64.74% and 74.81% of the terrain, respectively, and the class IV and V coefficients accounted for 86.40% and 92.69% of the terrain, respectively (Table 7). These results showed that the interpretative ability and degree of influence of FarmLS on surface temperature in the hills was stronger. In hills,

the class IV and V of ArtLS $R^2$ values accounted for 64.26% and 43.52% of the terrain, respectively. The percentage of the class V of ArtLS $R^2$ values in the plains was 56.20% and was focused on the surrounding areas of Beijing, Tianjin, and other cities, as well as on the junction of the hills and plains. In hills, the class IV and V coefficients accounted for 85.10% and 92.56% of the terrain, respectively (Table 7). This result showed that the interpretative ability of ArtLS to explain surface temperature was stronger in the hills and the plains, and the degree of influence of ArtLS on surface temperature was greater in the hills.

## DISCUSSION

### The distribution of hot spots

In this study, the distribution of hot spots obtained using the improved hot-spot analysis method was consistent with that obtained by the traditional method as a whole. In the areas with high topographic relief, we find slight differences, which was consistent with the expected results of the study. Therefore, if you want to study the aggregation of surface temperature or other attributes in a mountain city, or an urban agglomeration with a large difference in altitude, the improved three-dimensional hot-spot analysis method will theoretically more accurately reflect the clustering results of the study and be more in line with real life.

In the distribution of hot spots, other than the hot spots in Beijing and Tianjin, large hot spots appeared in Shijiazhuang, Handan, Xingtai, and Cangzhou, where the temperature was also high according to the statistical yearbook for the same year as the study. The higher surface temperature in Cangzhou was closely related to the development of industry, the higher surface temperature in Beijing and Tianjin was related not only to the high intensity of urbanization but also to the large population and large human-made heat emission. In other cities, the dimension was low, the landscape was dominated by FarmLS and ArtLS, the area of EcoLSP was small, the industrial development intensity was high, the artificial heat emission was relatively high, and the air pollution was serious, which was not conducive to heat diffusion. The region is northwest of the Taihang mountains and the slope was mostly from east to south (*Jiao et al., 2019*).

### Exploration of optimal analytical window granularity

On the basis of the 1,000-m-resolution LST data, we concluded that the optimal analytical window granularity for the expression of the relationship between landscape pattern and surface temperature was 5 × 5 km in the Beijing-Tianjin-Hebei urban agglomeration. On the basis of the 30-m-resolution surface temperature data, *Guo et al. (2012)* concluded that the optimal analytical window of the relationship between landscape pattern and surface temperature in the Pearl River Delta in China was 150 × 150 m. Conversely, *Weng, Liu & Lu (2007)* concluded that the optimal analytical window of the relationship between landscape pattern and surface temperature in Indianapolis, USA, was 120 × 120 m. *Lu et al. (2018)* used 120-m-resolution surface temperature data and found that the granularity of the analytical window of 1.8 × 1.8 km was optimal in Hangzhou, China.

This indicated that the granularity of the optimal analytical window for the expression of the relationship between landscape pattern and surface temperature was related not only to regional location and regional scale differences but also to the resolution of surface temperature and landscape pattern data sources.

## Interpretative ability of landscape pattern to explain surface temperature

In the Beijing-Tianjin-Hebei region, the interpretative ability of GraLS and WetLS to explain surface temperature was weak and inconsistent in different terrains. This phenomenon is not consistent with life experience and the results of *Kumar & Shekhar (2015)*, *Qiao, Tian & Xiao (2013)*. However, the study by *Yi, Hu & Li (2018)* found that the relationship between WetLS and surface temperature was unstable and possibly positively correlated comparing the relationship between WetLS and surface temperature in Jinan and Chongqing. Meanwhile, *Xiao et al. (2018)* compared the relationship among large GreenLSP, small GreenLSP, and surface temperature. They found that the cooling effect of large GreenLSP was more apparent and more stable, whereas the cooling effect of small GreenLSP was not obvious and was not stable. This finding indicated that in unit area, if the area percentage of a landscape was small, the relationship between the landscape pattern and surface temperature was unstable. In the Beijing-Tianjin-Hebei region, because WetLS is small and scattered, we should protect the Baiyangdian, Yuqiao reservoir and other wetland, maintain the flow of the river, and prevent the phenomenon of interruption. The area percentage of ForLS and EcoLSP increased by 10%, and the surface temperature dropped by 0.69 and 0.63 °C, respectively, indicating that the cooling capacity of EcoLSP, GraLS, and WetLS was lower than that of ForLS alone. It showed that the cooling capacity of GraLS and WetLS was less than that of ForLS and either one of the two was less than the cooling capacity of ForLS. According to the study by *Qiao, Tian & Xiao (2013)*, the cooling capacity of GraLS was smaller than that of ForLS because the thermal inertia of GraLS was smaller than that of ForLS, whereas the WetLS had a larger thermal inertia, and its cooling capacity was larger than that of ForLS. In fact, in the Beijing-Tianjin-Hebei region, a large area of WetLS concentrated in the low-altitude region of the Bohai Rim, the regional urbanization level was higher, the population was concentrated, industry had developed, and the sea land building and other human activities were evident, which all contributed to higher surface temperatures in the surrounding WetLS. ForLS, however, was distributed mainly in hilly regions with less human activities and a higher altitude, so it was reasonable to assume that the cooling effect of WetLS was less significant than that of ForLS. The NDVI value increased by 0.1 and the surface temperature dropped by 1.45 °C. This result indicated that the GreenLSP cooling effect was obvious, which was consistent with the research results of *Li et al. (2018)*, *Zhang, Odeh & Han (2009)*, *Chen et al. (2006)*, and *Huang, Cui & He (2018)*. GreenLSP not only can generate shadows and increase air humidity through transpiration (*Taha, 1997*; *De Munck et al., 2018*) but it also can increase surface roughness and improve the efficiency of air convection (*Gunawardena, Wells & Kershaw, 2017*) to achieve the purpose of cooling. The influence of ArtLS on surface temperature was twice as great as

that of FarmLS, and this phenomenon mainly related to the surface reflectance and the human activities (*Qiao, Tian & Xiao, 2013*).

Studies have shown that in regions with complex mountain environmental conditions, elevation, slope, slope direction, and other key topographic factors have a significant impact on surface temperature (*Jiao et al., 2019*). The interpretation and cooling capacity of EcoLSP on surface temperature in hilly areas was greater than that in the plains and the plateau. In hilly areas, the proportion of EcoLSP area was more than 70% (Table 3), the connectivity was strong, and the human disturbance was insignificant, and thus it had a strong cooling capacity. In the plains, the area percentage of EcoLSP was less than 10% (Table 3), the degree of EcoLSP fragmentation was high, the area percentage of ArtLS was large, the population was concentrated, and the artificial heat emission was large. The urban environmental stress made the forest vegetation growth worse, resulting in weak vegetation transpiration (*Philip & Azlin, 2005*). In the plateau area, although the area percentage of EcoLSP was 44.20%, GraLS was the main EcoLSP, which had a smaller cooling capacity than ForLS, and the area percentage of FarmLS was 48.77%. Therefore, the cooling capacity of EcoLSP in the plateau area was significantly lower than that in the hilly area. The interpretative ability and degree of influence of GreenLSP on surface temperature were stronger and larger in hilly areas. Although the NDVI value in the plains and the plateau was larger, the landscape in this region was predominately FarmLS (dry land) and ArtLS, both of which were thermal landscapes with higher surface temperatures. In hilly areas, which were in the northwestern study area, ForLS and GraLS were the primary landscapes, both non-thermal landscape, the altitude was higher, and the thermal inertia of FarmLS was significantly smaller than that of ForLS (*Qiao, Tian & Xiao, 2013*). The interpretative ability and degree of influence of FarmLS and ArtLS on surface temperature were stronger and greater in hilly areas. The main landscapes in hilly areas were EcoLSP and ForLS. If the area of ArtLS and FarmLS increased, which occupied the original EcoLSP, the EcoLSP area decreased. In addition, a lot of human activity not only increased the fragmentation of EcoLSP but also introduced environmental disturbances that could affect the growth of the surrounding forest and cooling capacity to reduce the cooling ability of EcoLSP (*Jones et al., 1990*; *Shen et al., 2015*).

However, we analyzed only the interpretative ability of landscape pattern to explain surface temperature from the perspective of an urban agglomeration scale. It did not conduct relevant research from the perspective of different urban areas and compare them. Therefore, a comparative analysis of differences in interpretative ability between the urban agglomeration scale and the city scale should be the primary focus of follow-up research. In the process of an interpretative ability analysis, to avoid the multicollinearity problem among landscape proportions, we adopted single-factor regression analysis. In the follow-up study, multifactor regression analysis could be considered to explore the interpretative ability of different landscapes to explain surface temperature.

## CONCLUSIONS

In this study, we quantitatively analyzed the spatial pattern characteristics of surface temperature and landscape using an improved three-dimensional hot-spot analysis

method. With the help of the Pearson correlation coefficient, OLS, and GWR model, we explored the optimal analytical window granularity of the relationship between landscape pattern and surface temperature, and we discussed the spatial heterogeneity of the degree of influence and the interpretative ability of four landscapes on surface temperature. The main conclusions follow.

1. The improved three-dimensional hot-spot analysis method is more suitable for the spatial heterogeneity analysis of surface temperature in some mountainous cities or urban agglomerations with complex topography and a large difference in altitude, and the results obtained are more in line with the real life scene in concept than the traditional two-dimensional hot-spot analysis method.

2. Under an analytical window granularity of $5 \times 5$ km, the Pearson correlation coefficient between ForLS, EcoLSP, GreenLSP, and surface temperature in the Beijing-Tianjin-Hebei urban agglomeration was larger than that obtained using other analytical window granularity values.

3. In the Beijing-Tianjin-Hebei region, the hot spots of surface temperature were distributed mainly in the southwestern plain, southeastern hills, Beijing, Tianjin, Shijiazhuang, and other urban areas, primarily those with ArtLS and FarmLS. The cold spots were distributed mainly in the northern hills and the Yanshan and Taihang mountains.

ForLS, FarmLS, ArtLS, GreenLSP, and EcoLSP had a strong ability to explain surface temperature, whereas GraLS and WetLS had a weak ability to explain surface temperature. There was spatial heterogeneity in the interpretative ability of EcoLSP, GreenLSP, ArtLS, and FarmLS to explain surface temperature. The interpretative ability and the thermal control ability of these four landscapes were more significant in the hills than in the plains and the plateau. The interpretative ability of EcoLSP to explain surface temperature was stronger at the junction of the plains and hills, especially in the Yanshan and Taihang mountains. The interpretative ability of ArtLS was stronger in the areas surrounding of cities, including Beijing and Tianjin, and at the junction of the plains and hills.

Therefore, under the background of coordinated and integrated development in the Beijing-Tianjin-Hebei urban agglomeration, to avoid a highly fragmented landscape, urban construction should be rationally distributed. The area of EcoLSP, ForLS within and between cities should be increased, especially around Beijing and Tianjin, and the level of development in the Yanshan and Taihang mountains and other areas dominated by natural landscape should be reduced. Doing so will promote the sustainable development of the Beijing-Tianjin-Hebei region.

## ACKNOWLEDGEMENTS

The authors would like to thank everyone who contributed to the study and research. We thank Accdon for its linguistic assistance during the preparation of this manuscript.

### Funding

This research was funded by the Major Project of the National Natural Science Foundation of China (No. 41590841), the National Key Research and Development Plan (No. 2016YFC0503001), the Natural Science Foundation of Tianjin, China (No. 18JCYBJC90900), the Project of Scientific Research Plan of Tianjin Education Commission (No. 2018KJ164), and Tianjin Education Commission Research Project (No. 2016CJ20). The funders had no role in study design, data collection and analysis, decision to publish, or preparation of the manuscript.

### Grant Disclosures

The following grant information was disclosed by the authors:
Major Project of the National Natural Science Foundation of China: 41590841.
National Key Research and Development Plan: 2016YFC0503001.
Natural Science Foundation of Tianjin, China: 18JCYBJC90900.
Project of Scientific Research Plan of Tianjin Education Commission: 2018KJ164.
Tianjin Education Commission Research Project: 2016CJ20.

### Competing Interests

The authors declare that they have no competing interests.

### Author Contributions

- Dongchuan Wang conceived and designed the experiments, performed the experiments, analyzed the data, contributed reagents/materials/analysis tools, authored or reviewed drafts of the paper, approved the final draft.
- Zhichao Sun conceived and designed the experiments, performed the experiments, analyzed the data, prepared figures and/or tables, authored or reviewed drafts of the paper, approved the final draft.
- Junhe Chen performed the experiments, analyzed the data, prepared figures and/or tables, authored or reviewed drafts of the paper, approved the final draft.
- Xiao Wang analyzed the data, contributed reagents/materials/analysis tools, prepared figures and/or tables, authored or reviewed drafts of the paper, approved the final draft.
- Xian Zhang analyzed the data, prepared figures and/or tables, authored or reviewed drafts of the paper, approved the final draft.
- Wei Zhang analyzed the data, contributed reagents/materials/analysis tools, prepared figures and/or tables, authored or reviewed drafts of the paper, approved the final draft.

### Data Availability

The raw measurements are available in the Supplemental Files. Remote-sensing data were downloaded from the Geographic Spatial Data Cloud and EARTHDATA Search.

## Supplemental Information

Supplemental information for this article can be found online at http://dx.doi.org/10.7717/peerj.7874#supplemental-information.

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
