# Peer review of "Analyzing the interpretative ability of landscape pattern to explain thermal environmental effects in the Beijing-Tianjin-Hebei urban agglomeration"

_PeerJ, doi:10.7717/peerj.7874_

## Round 0.1 · original submission · Major Revisions

Overall the manuscript represents a good contribution to science, but it needs some cleaning up and major revisions are in order to continue consideration. First, although the English is good, language, sentence conjugation and writing style all would benefit from a native speaking reviewing the final draft. I think this would benefit flow and readability. The reviewers have also brought up some points about experimental design and validity of findings, which if corrected would also greatly improve this manuscript. If you could fix the suggested changes, address the reviewers other suggestion and have this reviewed by an native English speaker it would benefit greatly and we would like to see a corrected version.

Reviewer 1 ·

Basic reporting

From my point of view, the English language used in all text is good (I am not a native speaker).
The topic of the paper is more relevant for scientific debate and the selected study area is quite an important example. However, I think that the introduction section does not contain all relevant literature. In my opinion there are many studies about the topic covered that the authors should insert for improve this section. The following references could be considered:
- Bonafoni, S., Baldinelli, G., Verducci, P., Presciutti, A., 2017. Remote sensing techniques for urban heating analysis: a case study of sustainable construction at district level. Sustain. For. 9 (8).https://doi.org/10.3390/su9081308 26 July 2017, Article number 1308. (ISSN: 20711050).
- De Munck, C., Lemonsu, A., Masson, V., Le Bras, J., Bonhomme, M., 2018. Evaluating the impacts of greening scenarios on thermal comfort and energy and water consumptions for adapting Paris city to climate change. Urban Clim. 23, 260–286. https://doi.org/10.1016/j.uclim.2017.01.003.
- Zullo F., Fazio G., Romano B., Marucci A., Fiorini L., 2019. Effects of Urban Growth Spatial Pattern (UGSP) on the Land Surface Temperature (LST): a study in the Po Valley (Italy). Sci Total Environ. 2019 Feb 10;650(Pt 2):1740-1751. doi: 10.1016/j.scitotenv.2018.09.331.
- Huang, M., Cui, P., He, X., 2018. Study of the cooling effects of urban green space in Harbin in terms of reducing the heat island effect. Sustain. For. 10 (4). https://doi.org/10.3390/su10041101.
Furthermore, the literature about both the urbanization phenomena does not updated. I suggest to the authors some important studies on this important topic:
- The Worldwatch Institute, 2007. State of theWorld, Our Urban Future. Norton, NY;
- Barrington-Leigh, C., Millard-Ballb, A., 2015. A century of sprawl in the United States. PNAS https://doi.org/10.1073/pnas.1504033112.
- Angel, S., Parent, J., Civco, D., 2012. The fragmentation of urban landscapes: global evidence of a key attribute of the spatial structure of cities, 1990–2000. Environ.Urban. 24, 249–283. http://dx.doi.org/10.1177/0956247811433536.
- European Commission, 2006. Urban Sprawl in Europe: The Ignored Challenge, EEA Report 10. p.60.
Furthermore, the authors should be explain the acronyms OLS and GWR (line 92). The study area could be completed by inserting both the urban (in %) and the population density (in inhab/km2).
Regarding the figures, I think that some of them could be improved. Here are some suggestions:
Figure 1: The authors should insert a hillshaded relief such as basemap layer for the study area region. Furthermore, the used label must be reorganized.
Figure 2: The used legend is not useful. The authors should insert a graduated color legend where they show the used altitude intervals for distinguished the plain, hill and plateau belts.

Experimental design

In my opinion, the research question does not clearly defined. For this reason, I think that the authors should rewrite the abstract part, also because they insert their findings through various acronyms that are described later in the paper. Therefore, the abstract should be rephrased describing only a brief introduction to the topic, the main aims of this work and how their findings can be used in the sustainable and integrated development in Beijing-Tianjin-Hebei urban agglomeration.
The paragraph Methods is quite articulated. I suggest to the authors to include a flowchart which can facilitate the reader in understanding this part. The flowchart can increase the clarity of the various phases that compose the methodology.

Validity of the findings

The applied methods are functional to the analysis carried out and they are supported both by a significant amount of data and statistical tests. However, I believe that the conclusions section must be improved. That is because the conclusion is not really a conclusion but a summary - first part just repeats some facts stated already below. State only, what is new, how you contributed to science and if there can anything else evolve from your findings (i.e. lines 383-388 and lines 488-496).

Additional comments

In general, I think that the manuscript is a nice study and it fit the aims of the journal. I consider this a solid case study paper and my suggestions could improve the manuscript. These are minor revisions that, once made, will allow the paper ready for to be published in the journal.

Reviewer 2 ·

Basic reporting

The manuscript "Analyzing the interpretative ability of landscape pattern to explain thermal environmental effects in the Beijing-Tianjin-Hebei urban agglomeration" studied the relationship between thermal environment and landscape pattern over the Beijing-Tianjin-Hebei Area. It's an interesting study but has a lot of flaws. It also suffers from a poor writing style. I think this manuscript is not appropriate for publication until significant improvements are made.

Experimental design

1) The authors used hot-spot analysis to study the clustering of land surface temperature (Ts). Ts is directly derived by the solar radiance and highly influenced by the land surface material, 3-dimension structure and other environmental factors. Not as socia-economical variables, What can be indicated from the the clustering of Ts? What's the meaning of the clustering of Ts?

2) How to determinate the landcape type for the window granularity. For example, for the window of 7*7, there were several landscapes in this window. How to determinate the lanscape of this window?

Validity of the findings

1) For Section 3.3.1 and 3.3.2, some results seem to be unreasonable. For example, the result indicated the negative relationship between the water body and Ts. However, it's well known that water body has obvious Cooling effect. This may be attributed to the simple method.

2) The discussion section mostly describes the results and lacks in-depth analysis. In addition, the intellectual merits of the result of this paper should also be discussed.

3) The 1st paragraph of the Section 4.3 looks like a literature review. The authors need to compare their results with other studies and discuss on it.

Additional comments

1) The literature review needs to be strengthened. A much better summary of the methods and variables employed in UHI pattern studies should be given.

2) The data section is too weak. The authors need to provide a much detailed description on data, not only in the Table. For example, what's the Ts data used in this paper? The method of the Ts product (MODIS) should be described. And the accuracy or reliability of this Ts product is better to be explained based on other references.

3) Sort the references.

4) L37, should be "between atmosphere and land surface"

5) L66-70, It's not true. There are a lot of UHI studies which introduced several environmental factors.

6) L73-75. It's also not true. There are a lot of UHI studies on metropolitan areas.

7) L220-221, Not clear

8) L223-226. Too detailed. Suggest to delete this description

9) L249, Please give the original or classic reference

10) L273-285, have little relationship with the topic of this paper.

·

Basic reporting

The English language throughout the manuscript could use considerable improvement to allow for increased readability among the journal’s international audience. Below are a few examples of where improvements could be made, though the entire manuscript would benefit from additional proofreading and review by a native English speaker.

1. Certain statements could use significant rewording to increase clarity, as their current phrasing makes comprehension difficult (e.g. Lines 12, 95, 478-481).

2. Other statements, while comprehensible, are vague and could use additional detail. For example, on Lines 55-56: the correlation was the “most significant” compared to what? On Line 95, what sort of “other aspects”?

3. “This study” generally does not perform any of the actions that carried out by the authors (e.g. Lines 85, 163, 226, 229). These instances could be replaced with “we”.

4. Sentences generally should not begin with conjunctions, such as “and” (e.g. Line 26, 77, 94). These could be removed.

Experimental design

1. While the main research gap appears to be laid out in Lines 66-70, your Introduction needs more detail leading up to your specific research questions. I suggest adding further justification for the objectives of the study that are outlined on Lines 84-97.

2. Lines 84-97: To increase the clarity and replicability of the study’s methods, methodological and analytical details mentioned in the Introduction would be more relevantly included in the Methodology section. For example, it could be stated that Pearson correlation coefficients were used to compare landscape pattern and surface temperature on Line 155, instead of the vague “relationship analysis”.

3. Lines 152-153: Additional clarity on methods would be helpful. How were the different landscapes integrated to produce EcoLSP and NDVI? Was NDVI not derived from the MODIS data, as described in Lines 127-129?

4. Lines 220-221: The claim that the different resolutions of basic data did not affect your research’s results does not have clear support. I would suggest providing some validation for this claim and/or reducing the certainty of the statement

Validity of the findings

1. Lines 304-306: How was it determined that 5 km x 5 km was a turning point at which r2 became strong? This claim was not clearly supported by Figure 5, which shows only slight peak at the 5 km x 5 km granularity for 2 of the 5 landscapes.

2. Lines 383-388: These statements make normative claims that appear to overstate the author’s findings, which are entirely correlative. Additionally, these statements would be more appropriately placed in the Discussion/Conclusion, if they are to be included in the paper at all.

3. Lines 508-510: As it is currently stated, this conclusion does not appear to be fully supported by the results that were shown (see above comment for Lines 304-306). If this conclusion specified that the 5 km x 5 km optimal granularity was observed for only the ForLS and EcoLSP landscape patterns, then that would greatly increase the validity of this finding.

Additional comments

1. Overall, results presented in this manuscript and methods used to produce them appear to be scientifically sound. The primary issues arise in the writing and in the interpretation of the results, both of which should be revised before acceptance. Care should particularly be taken to avoid attributing the relationships observed in the results to factors that were not explicitly explored in the study, as is done in the Discussion (e.g. Lines 444, 458)

2. The repeated use of “landscape” to refer to the different land cover types (ArtLS, ForLS, FarmLS) could lead to confusion among readers, as it does not align with many widely used definitions of a “landscape” (e.g. Forman & Godron 1981, Turner et al. 2001). Instead, I would suggest “land cover types” as a consistent substitute, as it is already used to refer to these metrics on Line 119.

---

## Round 0.2 · accepted · Accept

Thank you for addressing the reviewers comments, and for having the language reviewed. The manuscript appears to have greatly improved and we are happy to accept it at this time.

Reviewer 1 ·

Basic reporting

In my opinion, the literature references are now complete and the manuscript represent a good contribution to science.

Experimental design

I think that the authors have clear addressed all my suggestions. The paragraph Methods are now clear and the flowchart increase the clarity for the readers. This expedient allows the possibility to replicate.The authors have better specify their research question.

Validity of the findings

In my opinion, the conclusion section has been improved and it supported by the results. Furthermore, it links to research question.

Additional comments

Dear Authors,
thanks for the revision of article.
I found you have clearly addressed all points I have suggested in my previous review.
The only small suggestion is to control some typos (i.e. line 152) and puntuaction errors present in the text.
I would like to recommend the paper for publication.

Reviewer 2 ·

Basic reporting

no comment.

Experimental design

no comment.

Validity of the findings

no comment.

Additional comments

The revision made significant improvement and I'm satisfied with it.